# Comparison of Percutaneous Endoscopic Surgery and Traditional Anterior Open Surgery for Treating Lumbar Infectious Spondylitis

**DOI:** 10.3390/jcm8091356

**Published:** 2019-09-01

**Authors:** Tsai-Sheng Fu, Ying-Chih Wang, Tung-Yi Lin, Chia-Wei Chang, Chak-Bor Wong, Juin-Yih Su

**Affiliations:** Department of Orthopedic Surgery, Chang Gung Memorial Hospital, Keelung branch 20401, Taiwan and School of Medicine, Chang Gung University, Taoyuan 33305, Taiwan

**Keywords:** percutaneous endoscopy, spine infection, infectious spondylitis, anterior fusion surgery

## Abstract

Minimally invasive surgery is becoming popular for treating spinal disorders. The advantages of percutaneous endoscopic debridement and drainage (PEDD) for infectious spondylitis include direct observation of the lesion, direct pus drainage, and earlier pain relief. We retrospectively reviewed 37 patients who underwent PEDD and 31 who underwent traditional anterior open debridement and interbody fusion with bone grafting from 2004 to 2012. The causative organisms were isolated from 30 patients (81.1%) following PEDD, and from 25 patients (80.6%) following open surgery (*p* = 0.48). *Staphylococcus aureus* was the most common pathogen (38.2%). In the PEDD group, blood loss (<50 mL versus 585 ± 428 mL, *p* < 0.001) was significantly lesser and the duration of hospitalization (24.4 ± 12.5 days versus 31.5 ± 14.6 days, *p* = 0.03) was shorter than that in the open surgery group. Serologically, there were significantly faster C-reactive protein (CRP) and erythrocyte sedimentation rate (ESR) normalization rates in the PEDD group (*p* < 0.001, *p* = 0.009, respectively). In the two-year follow-up radiographs, 26 out of 30 (86.7%) open surgery patients showed bony fusions of the infected segments. On the contrary, sclerotic change of the destructive endplates was observed and the motion of infected spinal segments was still preserved in the PEDD group. There was no significant difference in the change of sagittal profile, including primary correction gain, correction loss, and actual correction gain/loss. PEDD is an effective alternative option and should be considered prior to traditional extensive spinal surgery—particularly for patients with early-stage spinal infection or serious complicated medical conditions.

## 1. Introduction

Spine infection accounts for about 2% to 4% of all bone infections [1]. The incidence of spinal infections has recently been increasing primarily due to prolonged life expectancy, improvements in diagnostic techniques, increasing iatrogenic spinal infections, intravenous drugs use, and the increased number of immunocompromised individuals. The overall incidence of infective spondylitis is estimated to be as high as 0.2 to 2.4 cases per 100,000 people [2]. The typical clinical symptoms include severe back pain, night pain, resting pain and fever with or without sciatica. The initial radiographic signs include a pathological fracture, a collapsed intervertebral disc space with or without erosion of the vertebral endplate or kyphosis.

Conservative treatment, comprising of long-term antibiotics, bed rest, and external mechanical support is the treatment of choice for the majority of the patients. Despite antibiotic treatment, additional surgery was required in 25% to 55% in these patients [3,4]. Surgical intervention must be considered in patients with significant bone destructions, mechanical instability or deformities, neurological impairments, epidural abscesses, failed conservative treatment or when biopsy data are needed [5,6,7].

Anterior spinal structures are affected in most cases of infectious spondylitis, whereas the posterior structures are normally unaffected. The disease spreads to multiple segments in more than 20% of the patients [8,9]. Traditionally, an anterior approach is normally selected for focus removal and placement of an autogenous or allogenous strut bone graft (ventral intersomatic fusion) [10,11]. To increase the stability of the affected spinal region and to maintain spinal alignment, some surgeons recommend additional posterior instrumentation (ventrodorsal fusion) [12,13]. Nevertheless, significant perioperative morbidities including donor site wound problems, failure of implants, dislodgement of bone graft, neurovascular injury, and anesthetic risks are particularly increased in the elderly patients or in those with multiple co-morbidities.

Nowadays, more spine surgeons choose minimally invasive endoscopy to treat infectious spondylitis due to increased familiarity and the improvement in endoscopy techniques [14,15,16]. In particular, percutaneous endoscopic debridement and drainage (PEDD) is performed under local anesthesia with a 0.5-inch wound to treat spine infection. This decreases the complications associated with general anesthesia, which may not be suitable for the elderly or critically ill patients. With PEDD, one can directly observe the lesion, collect sufficient specimen, drain pus directly, release intradiscal pressure, irrigate inflammatory factors—and most importantly—relieve pain immediately. The posterolateral endoscopic technique was used by [17] to treat 15 patients; all the patients showed an immediate reduction in back pain after surgery and the infections were subsequently successfully treated with parenteral antibiotic treatments. Yang and Fu used percutaneous endoscopy to identify the causative bacteria in 19 out of 21 (90.5 %) biopsy specimens from spondylitis patients with advanced lumbar infectious and successfully treated these patients without any surgical complications or neurological deficits [18]. Nevertheless, no literature has compared the outcomes of PEDD and traditional anterior open surgery. This study compares the clinical and radiographic outcomes of percutaneous endoscopic surgery and traditional anterior open surgery for treating infectious spondylitis.

## 2. Materials and Methods

We performed a retrospective review of 68 single-level lumbar infective spondylitis patients at the Chang Gung Memorial Hospital—a tertiary referral center—between 2004 to 2012. This retrospective study was approved by the institutional review board of Chang Gung Memorial Hospital (Institutional Review Board of Chang Gung Medical Foundation Reference Number: 103-6277B). Out of these, 37 patients underwent PEDD surgery and 31 received traditional anterior open surgery. Infectious spondylitis was diagnosed based on clinical symptoms, elevated erythrocyte sedimentation rate (ESR) and C-reactive protein (CRP) levels, radiographs, and magnetic resonance imaging results. All the patients presented with intractable back pain and required narcotics for pain control and bed rest. Those who had neurological deficits, previously undergone spine surgery, cauda equina syndrome, multiple levels of infection, severe local kyphosis (more than 30°), or large bony defects (more than half of the vertebrae body) were excluded. Patients with marked spinal instability, large bony defect, severe kyphosis, neurological deficits, or cauda equina syndrome may require extensive surgical decompression, stabilization and further reconstruction of the spinal structure. Parenterally sensitive antibiotics were administered to the patients after surgery—according to the culture reports. Intravenous antibiotics were administered to the patients for at least three weeks according to the improvement of clinical symptoms and laboratory data. If the symptoms and laboratory data improved as expected, the antibiotics were given orally for another 4 to 6 weeks. The mean duration of intravenous antibiotic treatment after surgery is 22.8 days in the PEDD group, and 27.3 days in the open surgery group.

### 2.1. Surgical Procedure in the PEDD Group

PEDD was performed via a posterolateral percutaneous approach using the Yeung Endoscopic Spinal System (Richard Wolf GmbH, Knittlingen, Germany) under local anesthesia and conscious sedation by an anesthesiologist. All the patients were positioned in a prone position. First, a spinal needle was inserted directly into the infected region under the guidance of intraoperative fluoroscopy. The abscess within the lesion was aspirated and sent for culture. The collected specimens were then subjected to aerobic and anaerobic cultures, tuberculosis culture and polymerase chain reaction, fungal culture, and histopathological examinations. After that, a guide wire was introduced into the disc space through the spinal needle and the needle was subsequently withdrawn. After creating a small stab wound incision (approximately 0.5 to 1 cm), a dilator and a cannulated sleeve were guided over the wire and passed sequentially into the disc space. After verifying the correct position of the sleeve tip by fluoroscope in two orthogonal planes, the tissue-cutting tool was inserted to harvest a biopsy specimen first. Discectomy forceps were inserted through the dilator to extract additional tissue from the infected disc under fluoroscopic monitoring. The necrotic disc material and parts of the destructive vertebral endplates of adjacent vertebrae were removed as much as possible for further aerobic and anaerobic cultures, tuberculosis culture, polymerase chain reaction, fungal culture, and histopathologic examinations. After biopsy and debridement, irrigation was performed using normal saline, after which the intradiscal lesion was endoscopically examined. Finally, a drainage tube (with a diameter of 3.2 mm) was inserted into the debrided disc space and connected to a negative-pressure pump (Hemovac; Zimmer, Dover, OH). The drainage tubes were left in place till the drainage stopped or declined to less than 10 mL/day for three consecutive days. After the operation, the patients were immediately allowed to walk while wearing a brace. Wearing a brace reduced the range of motion by more than 42% to 69%, and the stability of thoracolumbar spine increased significantly [19].

### 2.2. Surgical Procedure in the Open Surgery Group

For the anterior open surgery, a retroperitoneal approach was used in all the patients. All infected tissues, necrotic tissues, and disc material with the endplates were radically resected up to the healthy bleeding bone. The extent of debridement depended on the degree of infection, which was assessed based on preoperative MRI and the intraoperative findings. The intraoperative specimens were sent for aerobic and anaerobic cultures, tuberculosis culture and polymerase chain reaction, fungal culture, and histopathological examinations. After radical debridement, an autogenous tricortical iliac strut bone graft was measured to take the length of the disc space into account and was then firmly placed into position without internal fixation. To obtain solid fusion, the graft was placed in contact with viable bone both above and below. Then, the drainage tubes were left in place until the drainage stopped or declined to less than 50 mL/day for three consecutive days.

### 2.3. Evaluations of the Outcomes

The two groups (PEDD versus open surgery) were compared for clinical symptoms, serological tests (white blood count, CRP, and ESR), perioperative parameters (intraoperative blood loss, length of hospitalization after surgery), and radiographic results. We recorded the preoperative and postoperative ESR and CRP values regularly to monitor treatment up to three months after surgery. To record the lumbar sagittal alignment correction gained by surgery and the correction loss over time, preoperative, immediately postoperative, and postoperative two-year radiographs (current) were compared by measuring the sagittal angles of the infected spinal segment (Figure 1).

To objectively measure a potentially false sagittal positioning in the infected spine segment, the segmental lordotic angles were measured and compared with those reported by Vialle [20]. The difference between the segmental lordotic angles and the normal angle as reported by Vialle was the extent of the false segmental positioning in degrees, named the sagittal correction angle. Hypolordosis was defined as a false position of the sagittal profile in the direction of a kyphotic deformity. False positioning in the direction of kyphosis was denoted by a negative sign (−) and false positioning in the direction of lordosis was denoted by a positive sign (+). A postoperative minus (−) preoperative lordotic angle indicated primary correction gain after surgery; a postoperative minus (−) current lordotic angle indicated correction loss; and a current minus (−) preoperative lordotic angle indicated actual correction gain/loss.

### 2.4. Statistical Analysis

All statistical analyses were performed using the SPSS ver. 13.0 software (SPSS Inc, Chicago, IL, USA). Mann-Whitney U-test was used to compare continuous variables, and the Fisher’s exact test was used for dichotomous variables. Statistical significance was set at a *p*-value of < 0.05. All the patients were followed up for at least two years after undergoing the surgery.

## 3. Results

We retrospectively enrolled 68 patients with infectious spondylitis who underwent surgical intervention; 37 and 31 of whom underwent PEDD and traditional anterior open surgery, respectively. There were 48 males and 20 females, and the average age was 57.8 years old (range: 38–82). Males were predominant in the spine infection study. There was no statistical difference in patient age, gender, and co-morbidities between these two groups (Table 1). The infected spinal regions were L1–L2 in two patients, L2–L3 in 13 patients, L3–L4 in 13 patients, L4–L5 in 24 patients, and L5–S1 in 16 patients. Most of the infections occurred in the L4–L5 level in both of the groups.

### 3.1. Clinical Results

During the PEDD, the volume of blood loss was <50 mL—which was minimal. The blood loss in the PEDD group was significantly less than that for the traditional anterior open surgery group (<50 mL versus 585 ± 428 mL, *p* < 0.001) (Table 2). Furthermore, the duration of hospitalization in the PEDD group was shorter than the open group (24.4 ± 12.5 days versus 31.5 ± 14.6 days, *p* = 0.03). For the patients who underwent PEDD, the time taken for CRP to return to normal (<5 mg/L) was 19.1 ± 10.7 days, and the time taken for ESR to return to normal (<20 mm/hour) was 38.4 ± 21.6 days (Table 2). In the traditional open surgery group, the time taken for CRP to return to normal (<5 mg/L) was 33.8 ± 17.9 days, and the time taken for ESR to return to normal (<20 mm/hour) was 50.8 ± 29.3 days. Serologically, there was a statistically significant faster CRP and ESR normalization rate in the PEDD group (*p* < 0.001, *p* = 0.009, respectively).

Out of the 37 patients who underwent PEDD, five who needed further open surgery—three with uncontrolled infection, and two with persisting back pain—were excluded from parameter analysis. There were no major complications following PEDD, except for two who had transient paresthesia in the affected lumbar segments. In the traditional open surgery groups, four patients had transient paresthesia, two had wound infections over the autogenous iliac bone donor site, and one with end-stage renal failure and fungal infection died due to poor control of infection. Four patients needed a repeat open surgery; two due to uncontrolled infections, and two due to dislodged bone grafts. Totally, 32 patients in PEDD group and 26 patients in open surgery group were taken into parameter analysis.

### 3.2. Radiographical Results

Radiographically, the primary correction gain after surgery was slightly higher in the open surgery group (Table 2). However, there was no significant difference in the change of sagittal profile, including primary correction gain (2.20° ± 2.91° versus 3.04° ± 3.65°), correction loss (2.41° ± 3.92° versus 3.68° ± 2.76°), and actual correction gain/loss (−0.21° ± 5.59° versus −0.64° ± 5.01°). In the follow-up radiographs, the successful bony fusions of the infected segments appeared in most of the anterior open surgery cases (Figure 1). A successful bone fusion was determined by continuously trabecular bony bridging between vertebrae and bone graft. There was no change in the sagittal angle of the targeted spinal segment between flexion-extension dynamic plain radiographic views because not every patient would consent to computerized tomography scans. Twenty-six out of 30 (86.7%) open surgery patients showed solid fusion at two years for the follow up. Nevertheless, 8 patients had obvious subsidence of the bone grafts due to damaged endplates in the early periods. Conversely, the destructive endplates of the infected segments became sclerotic, although decreased intervertebral disc height and local kyphosis were observed after successful treatment in the PEDD group (Figure 2). However, the motion of these infected spinal segments was still preserved after successful PEDD treatment. In the longer follow up, two patients showed spontaneous fusion in the PEDD group (Figure 2).

### 3.3. Microbiological Results

We successfully isolated the causative organism from 30 patients (81.1%) following PEDD and from 25 patients (80.6%) following the open surgery (*p* = 0.48) (Table 3). The positive culture rate of PEDD was comparable to that of traditional open surgery. Totally, 21 patients were infected with *Staphylococcus aureus* (11 with Methicillin-resistant strain and 10 with Methicillin-sensitive strain), eight with *Mycobacterium tuberculosis*, nine with *Enterococcus* species, six with *Streptococcus* species, three with gram-positive bacilli, two with *Pseudomonas aeruginosa*, three with fungal infection, and the other three with *Klebsiella pneumoniae*, *Peptostreptococcus micros*, and *Stenotrophomonas maltophilia*. Among the 55 cultures that exhibited growth, *Staphylococcus aureus*—which typically exists on human skin—was the most common pathogen (38.2%). A regimen consisting of systemic antibiotics, antituberculous agents, or antifungal agents was administered according to sensitivity studies for each identified pathogen.

## 4. Discussion

Percutaneous endoscopic surgery is gaining popularity in the treatment of spinal disorders. Most patients would prefer to undergo a minimally invasive spinal surgery rather than a traditionally open surgery. Percutaneous endoscopy can be performed for degenerative spine disorders as well as infectious spondylitis patients. Previous studies have not compared the outcomes between PEDD and traditional anterior open surgery for lumbar infectious spondylitis. The results of this study showed that patients who underwent PEDD have a smaller blood loss and a shorter duration of hospitalization without significant perioperative morbidities. The positive culture rate of PEDD is comparable to that of traditional open surgery (81.1% versus 80.6%, *p* = 0.48). Radiographically, there was no significant difference between the PEDD and traditional open surgery groups. PEDD cannot replace the “gold standard” place of anterior open surgery, but it provides a good alternative option to treat lumbar infectious spondylitis. PEDD is good and suitable for earlier stage infectious spondylodiscitis; patients who are not suitable for general anesthesia such as the elderly or very-ill; patients who need a earlier diagnosis or detection of the causative organisms; and to preserve a better motion of the spine. It may not be suitable for patients with marked spinal instability, large bony defect, severe kyphosis, and neurological deficits, as these patients may require extensive surgical decompression, secure stabilization, and further reconstruction of the spinal structure.

Infectious spondylitis is essentially a medical disease. Literature has shown that uncomplicated spondylodiscitis can be adequately treated by early antibiotic therapy and immobilization [3,4]. The descriptive study by Karadimas concluded that nonoperative treatment was effective in 90% of the patients [21]. A recent randomized controlled trial by Bernard compared the six to 12 weeks of parenteral antibiotic treatment and showed that the therapy can be safely shortened to a total of six weeks without increasing the risk of relapse, failure, and infection-related mortality [22]. The high cure rate was related to early diagnosis and proper antibiotic treatment that allowed a lesser progression of the infectious process and a better response to therapy. However, there was a difference of opinion and the conclusion was that along with antibiotics treatment, additional surgery is required in 25% to 55 % of patients with infectious spondylodiscitis and that the relapse rate is 2% to 4 % when treated with antibiotics only [3,4].

Although using sensitive parenteral antibiotics plays the main role in the treatment of spine infection, surgical intervention must be considered in patients with mechanical instability or deformities, neurological impairments, epidural abscesses, failed conservative treatment, or when biopsy data are needed. The selection and timing for using antibiotics to treat spine infection are very important. Pre-biopsy empirical antibiotics may affect the culture results of the causative organism and bother further successful infection control. Wang and Fu identified the causative bacteria in 32 out of 41 infectious spondylitis (78.0%) cases via the PEDD and showed that prebiopsy empirical antibiotic therapy is associated with a lower positive culture rate and increased the need for subsequent open surgery [16]. The patients with positive cultures were more likely to have initially adequate sensitive parenteral antibiotic treatment, which resulted in successful infection control and better clinical outcomes.

Currently, many different surgical methods are being used for the treatment of infectious spondylodiscitis. Traditionally, anterior open debridement of infected bone and soft tissue with interbody fusion with autogenous or allogenous bone graft with or without instrumentation for mechanical instability is the gold standard of surgical treatment. Anterior column reconstruction can be carried out with either a structural graft or a cage. There is controversy regarding the use of metal implants in active infection, even though several literatures have demonstrated that the use of metallic implants in an infected area of the spine does not lead to persistence or recurrence of the infection [23,24]. Besides, metal implants may hinder healing of the infection. Therefore, an increased rate of septic loosening was discussed and we chose an autologous tricortical iliac structural allograft as the bone graft. Anterior fusion alone is appropriate for patients with single-level spondylodiscitis and minor substance loss—especially in the lumbar spine [25]. An additional posterior instrumentation is indicated for multiple-level spondylodiscitis and extensive deformity. Linhardt reported that isolated anterior open surgery has less pain and statistically significant better clinical outcomes than combined anterior surgery with posterior instrumentation group [26]. The anterior approach provides direct access and improved exposure to the most commonly affected parts of the spine. Nevertheless, significant perioperative morbidities—including wound problems, failure of implants, dislodgement of bone grafts, neurovascular injuries, and anesthetic risks—are particularly increased in the elderly patients and those with multiple co-morbidities.

Percutaneous endoscopic discectomy (PED) was first employed for the treatment of uncomplicated herniated discs in the early 1980s. The clinical outcomes of these procedures are comparable to those of conventional open surgery. The minimal invasiveness and simplicity of the technique led surgeons to use PED for spinal debridement and drainage and as a modality for the treatment of earlier-stage infectious spondylodiscitis. The same posterolateral endoscopic technique was used by [17] to treat 15 patients with pyogenic spondylodiscitis; all patients showed immediate back pain reduction after surgery and the infections were successfully treated with subsequent parenteral antibiotic treatment. Yang and Fu reported that the causative bacteria were identified in 28 out of 32 (87.5%) spine infection patients; 27 (84.4%) patients had satisfactory relief of their back pain after PEDD; and 26 (81.3%) patients recovered uneventfully after PEDD and sequential antibiotic therapy [27]. No major surgery-related complications were found—except for three patients with transient paresthesia in the affected lumbar segment. Additionally, Mao [28] performed a systematic review and meta-analysis which included nine single-arm PEDD articles (158 patients). The pooled event rate was 82% (95% CI: 75% to 88%) for positive bacteria culture, 81% (95% CI: 73% to 87%) for pain control satisfaction, and 21% (95% CI: 15% to 29%) for reoperation [28]. Only a few studies have included transient paresthesia in the affected lumbar segment and local kyphosis.

PEDD is a minimally invasive and safe procedure for the treatment of lumbar spine infection—especially in earlier-stage infectious spondylodiscitis. It allows for earlier diagnosis, detection of the causative organisms, and provides accurate therapeutic treatment with identification of the sensitive antibiotics. In the current study, the clinical and radiographical results of the PEDD are comparable to that of traditional open surgery. There was no significant difference in the change of sagittal profile, including primary correction gain, correction loss, and actual correction gain/loss between these two surgical methods. When compared to the anterior debridement and fusion surgery, the PEDD technique does not require the fusion of the infected vertebral segments and still preserves the mobility of the lumbar spine after infection control. It is a truly motion-preserving therapeutic technique for the treatment of spine infection. Furthermore, additional advantages of the PEDD include less blood loss, earlier recovery and pain control, earlier normalized CRP and ESR levels, and shorter duration of hospitalization when compared to traditional open surgery. PEDD is an effective alternative option and should be considered prior to traditional extensive spinal surgery—particularly for patients with an early-stage spinal infection. There are some limitations in our study: our cases were retrospective and not randomized, there was a selection bias by different surgeons whom chose different surgical methods, and the sample size was relatively small. Further studies with large samples need to be conducted.

## 5. Conclusions

Percutaneous endoscopic debridement and drainage (PEDD) has a good therapeutic and diagnostic value and is comparable to traditional open surgery for the treatment of lumbar infectious spondylitis. PEDD has lesser blood loss during operation, shorter duration of hospitalization, and is without significant perioperative morbidities—it is therefore a good alternative option to treat lumbar infectious spondylitis.

## Figures and Tables

**Figure 1 jcm-08-01356-f001:**
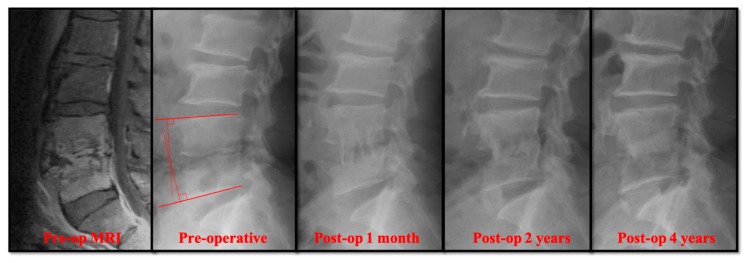
A 55-year-old man presented with severe pain in the lower back and was diagnosed with L4–L5 *Escherichia coli* infection. Four years after the traditional anterior open surgery, bony fusion was seen in the infected L4–L5 segment. The sagittal angle was measured by the angle of the perpendicular lines from the upper and the lower endplates.

**Figure 2 jcm-08-01356-f002:**
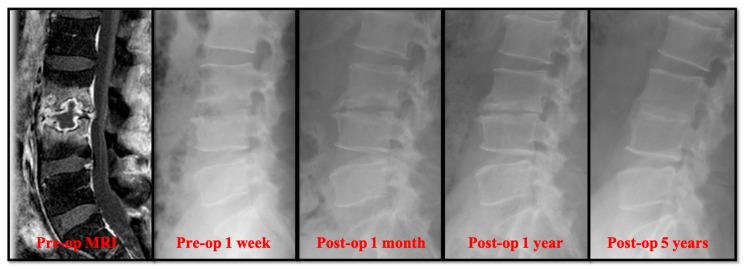
A 36-year-old man presented with severe pain in the lower back and leg. He was diagnosed with L3–L4 methicillin-resistant *Staphylococcus aureus* infection. The destructive endplate of the infected L3–L4 segment became sclerotic change although decreased intervertebral disc height and local kyphosis were observed after successful treatment in the PEDD group.

**Table 1 jcm-08-01356-t001:** Patients’ characteristics.

Characteristics	Total (*n* = 68)	PEDD (*n* = 37)	Open (*n* = 31)	*p*-Value
Mean age (years)	57.8 ± 13.4	56.5 ± 14.4	59.9 ± 12.1	0.15
Gender (Female/Male)	20/48	10/27	10/21	0.32
**Location of Infection**				
L1–L2	2	1	1	
L2–L3	13	6	7	
L3–L4	13	5	8	
L4–L5	24	14	10	
L5–S1	16	11	5	
**Co-Morbidity**				
Liver cirrhosis	16.1% (11/68)	13.5% (5/37)	19.4% (6/31)	0.26
Diabetes mellitus	14.7% (10/68)	16.2% (6/37)	12.9% (4/31)	0.35
Chronic renal failure	127.6% (12/68)	13.5% (5/37)	22.6% (7/31)	0.17
Coronary heart disease	8.8% (6/68)	8.1% (3/37)	9.7% (3/31)	0.41

**Table 2 jcm-08-01356-t002:** Clinical and radiographical outcomes.

Characteristics	PEDD (*n* = 37)	Open Surgery (*n* = 31)	*p*-Value
CRP return to normal (day)	19.1 ± 10.7	33.8 ± 17.9	<0.001 *
ESR return to normal (day)	38.4 ± 21.6	50.8 ± 29.3	0.009 *
Blood loss during surgery (mL)	<50	585 ± 428	<0.001 *
Duration of hospitalization (day)	24.4 ± 12.5	31.5 ± 14.6	0.03 *
Culture rate	81.1% (30/37)	80.6% (25/31)	0.48
Preoperative sagittal correction angle	−4.84° ± 8.68°	−7.58° ± 8.21°	
Postoperative sagittal correction angle	−2.64° ± 7.44°	−4.54° ± 8.32°	
Current sagittal correction angle	−5.05° ± 7.84°	−8.22° ± 8.09°	
Postop-preop lordotic angle	2.20° ± 2.91°	3.04° ± 3.65°	0.35
Postop-current lordotic angle	2.41° ± 3.92°	3.68° ± 2.76°	0.17
Current-preop lordotic angle	−0.21° ± 5.59°	−0.64° ± 5.01°	0.76

* Statistically significant difference (*p*-value < 0.05).

**Table 3 jcm-08-01356-t003:** Summary of causative microorganisms.

	PEDD Group	OPEN Group	Total
Methicillin-resistant *Staphylococcus aureus*	5	6	11
Methicillin-sensitive *Staphylococcus aureus*	6	4	10
*Streptococcus* spp.	4	2	6
*Enterococcus* spp.	6	3	9
*Pseudomonas aeruginosa*	1	1	2
*Klebsiella pneumoniae*	1	0	1
*Stenotrophomonas maltophilia*	0	1	1
*Peptostreptococcus micros*	0	1	1
G+ bacilli	2	1	3
*Mycobacterium tuberculosis*	4	4	8
Fungus	1	2	3
No growth	7	6	13

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
