# Peer review of "Comparison of Percutaneous Endoscopic Surgery and Traditional Anterior Open Surgery for Treating Lumbar Infectious Spondylitis"

_jcm, 2019, doi:10.3390/jcm8091356_

Round 1

Reviewer 1 Report

Fu et al. provide a retrospective review of 68 patients comparing PEDD and anterior open surgery.

Pathogens, intraoperative blood loss, hospitalization and CRP as well as ESR normalization rates were compared and PEDD was favourable regarding blood loss, hospitalization time and normalization rates of CRP and ESR.

Abstract:

The two-year follow up is an important part of your study, why not mention it in the abstract?

Introduction: 

Line 55: Why is PEDD not sufficient for elderly patients? Do these patients benefit even more from minimally invasive surgery?

Materials and Methods:

Line 73: Please justify the underlying reasons for the mentioned exclusion criteria.

What was the mean duration of antibiotic treatment after surgery?

line 88: How was debridement performed?

line 93: Please give a rationale for bracing (consider i.e. PMID: 23644685)

line 100: Please give a rationale for the tricortical iliac strut bone graft; why was no cage used?

Please add the ethics committee number.

Results:

line 180: "successful bone fusion": How was bony fusion measured? Did you use any radiographic scores?

Discussion:

line 223: Could you add a "perfect" indication for PEDD?

Author Response

JCM-582815

Comparison of Percutaneous Endoscopic Surgery and Traditional Anterior Open Surgery for Treating Lumbar Infectious Spondylitis

Dear Reviewer:

Thanks for providing us with the opportunity to respond to your expert comments of our manuscript entitled "Comparison of Percutaneous Endoscopic Surgery and Traditional Anterior Open Surgery for Treating Lumbar Infectious Spondylitis " and permitting us to revise the manuscript. In this revised manuscript, we try our best to answer the questions and comments. We have also made some modifications in the revised manuscript according to your opinions.

Reviewer #1:

The two-year follow up is an important part of your study, why not mention it in the abstract?

Reply: We totally agree with this comment of reviewer. We add the two-year follow up results to abstract in page 1 line 21, as “(p<0.001, p=0.009, respectively). In the two-year follow-up radiographs, 26 out of 30 (86.7%) open surgery patients showed bony fusions of the infected segments. On the contrary, sclerotic change of the destructive endplates was observed and the motion of infected spinal segments was still preserved in the PEDD group. There was no significant difference in the change of sagittal profile,”

Line 55: Why is PEDD not sufficient for elderly patients? Do these patients benefit even more from minimally invasive surgery?

Reply: Thank you for your correction. The expression of sentence is not good. PEDD is sufficient and suitable for elderly and ill patients. These patients get more benefits from minimal invasive surgery. We revise the sentence in page 2 line 58, as “This decreases the complications associated with general anesthesia that may not suitable for the elderly or critically ill patients.”

Line 73: Please justify the underlying reasons for the mentioned exclusion criteria.

Reply: This is a very good point. We add more description in page 2 line 82, as “ In those patients with marked spinal instability, large bony defect, severe kyphosis, neurological deficits, or cauda equina syndrome may require extensive surgical decompression, stabilization and further reconstruction of the spinal structure.”

What was the mean duration of antibiotic treatment after surgery?

Reply: As your suggestion, we add the description in page 2 line 88, as “ The mean duration of intravenous antibiotic treatment after surgery is 22.8 days in PEDD group, and 27.3 days in open surgery group.”

line 88: How was debridement performed?

Reply: As suggestion, we add the description in page 3 line 99, as “After creating a small stab wound incision (approximately 0.5 to1 cm), a dilator and a cannulated sleeve were guided over the wire and passed sequentially into the disc space. After verifying the correct position of the sleeve tip by fluoroscope in two orthogonal planes, the tissue-cutting tool was inserted to harvest a biopsy specimen first. Discectomy forceps were inserted through the dilator to extract additional tissue from the infected disc under fluoroscopic monitoring. The necrotic disc material and parts of the destructive vertebral endplates of adjacent vertebrae were removed as possible for further aerobic and anaerobic cultures, tuberculosis culture, polymerase chain reaction, fungal culture, and histopathologic examinations.”

line 93: Please give a rationale for bracing (consider i.e. PMID: 23644685)

Reply: Thank you for your suggestion. We add the description in page 3 line 112, as “Wearing brace reduced the range of motion by more than 42% and up to 69%, and the stability of thoracolumbar spine increased significantly.”

Kienle, A.; Sali S.; Michael O. Effect of 2 different thoracolumbar orthoses on the stability of the spine during various body movements. Spine 2013, 38, e1082-e1089.

line 100: Please give a rationale for the tricortical iliac strut bone graft; why was no cage used?

Reply: This is a very good question. We add the description in “Discussion section” page 8 line 279, as “Anterior column reconstruction can be carried out with either a structural graft or a cage. There is controversy regarding the use of metal implants in active infection, although several literatures have demonstrated that the use of metallic implants in an infected area of the spine does not lead to persistence or recurrence of the infection. Besides, metal implants may hinder healing of the infection; an increased rate of septic loosening was discussed. Therefore, we choose an autologous tricortical iliac structural allograft as the bone graft.”

Liljenqvist, U.; Lerner, T.; Bullmann, V.; Hackenberg, L.; Halm, H.; Winkelmann, W. Titanium cages in the surgical treatment of severe vertebral osteomyelitis. Euro Spine J 2003, 12, 606-612.

Ruf, M.; Stoltze, D.; Merk, H. R.; Ames, M.; Harms, J. Treatment of vertebral osteomyelitis by radical debridement and stabilization using titanium mesh cages. Spine, 2007, 32, e275-e280.

Please add the ethics committee number.

Reply: As your suggestion, we add the description in page 2 line 73, as “This retrospective study was approved by the institutional review board of Chang Gung Memorial Hospital (Institutional Review Board of Chang Gung Medical Foundation Reference Number: 103-6277B).”

line 180: "successful bone fusion": How was bony fusion measured? Did you use any radiographic scores?

Reply: This is a very good question. We determined “successful bone fusion” from radiograph by two points. First, it must present with continuously trabecular bone bridging between vertebrae and bone graft. Second, there is no change in sagittal angle of targeted spinal segment between flexion-extension dynamic plain radiographic views because not every patient would consent to computerized tomography (CT) scans. We add the description in page 6 line 207, as “A successful bone fusion was determined by continuously trabecular bony bridging between vertebrae and bone graft, and there is no change in sagittal angle of targeted spinal segment between flexion-extension dynamic plain radiographic views because not every patient would consent to computerized tomography scans.”

line 223: Could you add a "perfect" indication for PEDD?

Reply: As your suggestion, we add the description in page 7 line 248, as “PEDD is good and suitable for earlier stage infectious spondylodiscitis; elderly or very-ill patients, who is not suitable for general anesthesia; patients who need earlier diagnosis or detection of the causative organisms; preserve better motion of spine. It may not suitable for patients with marked spinal instability, large bony defect, severe kyphosis, and neurological deficits. Because those patients may require extensive surgical decompression, secure stabilization, and further reconstruction of the spinal structure.”

We appreciate the reviewer’s excellent comments, which have improved our revised manuscript. Thank you very much for your expert comments.

Warm regards,

Sincerely,

The authors

Reviewer 2 Report

This is a very interesting retrospecitve case series.  I look forward to reading future prospecitve studies comparing PEDD to open washout.  Thank you for your contribution to the literature.  

Author Response

JCM-582815

Comparison of Percutaneous Endoscopic Surgery and Traditional Anterior Open Surgery for Treating Lumbar Infectious Spondylitis

Reviewer #2:

This is a very interesting retrospecitve case series. I look forward to reading future prospecitve studies comparing PEDD to open washout. Thank you for your contribution to the literature.

Dear Reviewer:

Thanks for providing us with the opportunity to respond to your expert comments of our manuscript entitled "Comparison of Percutaneous Endoscopic Surgery and Traditional Anterior Open Surgery for Treating Lumbar Infectious Spondylitis " and permitting us to revise the manuscript. In this revised manuscript, we try our best to answer the questions and comments. We have also made some modifications in the revised manuscript. We will keep effort on further prospective study of PEDD as you suggested. Thank you for your opinion and interests.

We appreciate the reviewer’s excellent comments, which have improved our revised manuscript. Thank you very much for your expert comments.

Warm regards,

Sincerely,

The authors

Reviewer 3 Report

Authors showed successful treatment of 37 infectious spondylodiscitis cases with using PEDD and comparred the clinical results with using conventional anterior open approach.

The clinical results of PEDD was superior in the resurlts, indicating the possibility op MIS surgery for infectious spondylodiscitis.

I have some concerns about this manuscript.

There is no mention about the selective bias. How the surgeons selected the two surgical methods to treat the infection? In the past, they preferred open surgery and recently have they treated the patients with PEDD? or Was there any other reasons? Clinical result was estimated using CRP of 5.0mg/dl. This value seems too high to think that the infection is calming down. Authors have to add the information about more critical CRP value (such as 1.0mg/dl) and compare between two groups. What is the rate of spontaneous fusion in PEDD group at final follow-up? Was the follow-up rate 100%? or did the author select the patients who could be followed up? 

Author Response

JCM-582815 Comparison of Percutaneous Endoscopic Surgery and Traditional Anterior Open Surgery for Treating Lumbar Infectious Spondylitis

Dear Reviewer:

Thanks for providing us with the opportunity to respond to your expert comments of our manuscript entitled "Comparison of Percutaneous Endoscopic Surgery and Traditional Anterior Open Surgery for Treating Lumbar Infectious Spondylitis " and permitting us to revise the manuscript. In this revised manuscript, we try our best to answer the questions and comments. We have also made some modifications in the revised manuscript according to your opinions.

Reviewer #3:

There is no mention about the selective bias. How the surgeons selected the two surgical methods to treat the infection?

Reply: This is one of the limitations in this article. Since the current study is retrospective and not randomized. These 68 patients were treated by different surgeons in our institution. These surgeons may choose different methods according to their favorite and familiarity of surgical methods, the medical condition of patients, and the willingness from patients. We add the description in page 9 line 323 as “There are some limitations in our study, our cases were retrospective and not randomized, exists selection bias by different surgeons chosen different surgical methods, and the sample size was relatively small. Further studies with large samples need to be conducted.”

In the past, they preferred open surgery and recently have they treated the patients with PEDD? or Was there any other reasons?

Reply: This is a very good question. PEDD was performed under local anesthesia; prevent the complication of general anesthesia, with a small surgical wound, shorter hospitalization duration, and early ambulation. We add several reasons to explain it in page 2 line 55 as ” Nowadays, more spine surgeons choose minimally invasive endoscopy to treat infectious spondylitis due to increased familiarity and the improvement in endoscopy techniques [14-16]. In particular, percutaneous endoscopic debridement and drainage (PEDD) is performed under local anesthesia with a 0.5-inch wound to treat spine infection. This decreases the complications associated with general anesthesia that may not suitable for the elderly or critically ill patients. With PEDD, we can directly observe the lesion, collect sufficient specimen, drain pus directly, release intradiscal pressure, irrigate inflammatory factors, and most importantly, relieve pain immediately.”

Clinical result was estimated using CRP of 5.0mg/dl. This value seems too high to think that the infection is calming down. Authors have to add the information about more critical CRP value (such as 1.0mg/dl) and compare between two groups.

Reply: Thank you for your correction. We had written the wrong unit. In our institution, the normal range of CRP is <5 mg/L. (1mg/dL = 10 mg/L). We corrected the sentence in page 5 line 182 and 184 as “CRP to return to normal (<5 mg/L)”.

What is the rate of spontaneous fusion in PEDD group at final follow-up?

Reply: As suggestion, we add the rate of spontaneous fusion in PEDD group in page 6 line 215 as “However, the motion of these infected spinal segments was still preserved after successful PEDD treatment. In the longer follow up, there are 2 patients showed spontaneous fusion in the PEDD group.”

Was the follow-up rate 100%? or did the author select the patients who could be followed up?

Reply: No. In PEDD group, 5 who needed further open surgery, 3 with uncontrolled infection, and 2 with persisting back pain were excluded from parameter analysis. In open surgery group, 1 with end-stage renal failure and fungal infection died due to poor control of infection, and 4 patients needed a repeat open surgery were excluded from parameter analysis. We add more description in page 6 line 198 as “Totally, 32 patients in PEDD group and 26 patients in open surgery group were taken into parameter analysis.”

We appreciate the reviewer’s excellent comments, which have improved our revised manuscript. Thank you very much for your expert comments.

Warm regards,

Sincerely,

The authors
